# On the Design and Implementation of the External Data Integrity Tracking and Verification System for Stream Computing System in IoT [note 1]

**DOI:** 10.3390/s22176496

**Published:** 2022-08-29

**Authors:** Hongyuan Wang, Baokai Zu, Wanting Zhu, Yafang Li, Jingbang Wu

**Affiliations:** 1Faculty of Information Technology, Beijing University of Technology, Beijing 100124, China; 2School of Computer Science and Engineering, Beijing Technology and Business University, Beijing 100048, China

**Keywords:** data integrity verification, stream computing, internet of things, integrity tracking and verification system

## Abstract

Data integrity is a prerequisite for ensuring data availability of IoT data and has received extensive attention in the field of IoT big data security. Stream computing systems are widely used in the field of IoT for real-time data acquisition and computing. However, the real-time, volatility, suddenness, and disorder of stream data make data integrity verification difficult. According to the survey, there is no mature and universal solution. To solve this issue, we constructed a data integrity verification algorithm scheme of the stream computing system (S-DIV) by utilizing homomorphic message authentication code and pseudo-random function security assumption. Furthermore, based on S-DIV, an external data integrity tracking and verification system is constructed to track and analyze the message data stream in real time. By verifying the data integrity of message during the whole life cycle, the problem of data corruption or data loss can be found in time, and error alarm and message recovery can be actively implemented. Then, we conduct the formal security analysis under the standard model and, finally, implement the S-DIV scheme in simulation environment. Experimental results show that the scheme can guarantee data integrity in an acceptable time without affecting the efficiency of the original system.

## 1. Introduction

The rapid development of emerging technologies and applications such as the Internet of Things and 5G networks has led to a worldwide data explosion, which pushes human society into the era of big data. Due to the new computing and storage model of IoT big data, its management and protection measures are somewhat different from those of ordinary data; therefore, how to maintain and guarantee the integrity and consistency of data throughout its life cycle in the IoT big data environment has become one of the important issues in the field of data security [1].

Currently, batch offline computing and stream real-time computing are the main computing models for IoT big data [2]: batch computing is mainly a computing model for static persistent data, which usually stores the data first and then distributes the data and computing logic to distributed computing nodes for data computation—a common batch computing architecture is Hadoop; stream computing is mainly a computing model for data stream, it does not store all the data, but instead performs data computation directly in memory for a certain period of time.

Data integrity is a prerequisite for ensuring the availability of IoT data and is the key to the secure operation of IoT big data systems. According to the research, almost all current data integrity verification schemes are studied under the batch computing mode with a high degree of technical and research maturity, while there is no perfect solution for the data stream integrity problem under the stream computing mode. In the traditional batch computing model, data integrity verification mechanisms are divided into two categories based on whether fault-tolerant recovery measures are applied: provable data possession (PDP) and proof of retrievability (POR). However, the current PDP and POR schemes cannot be directly applied to IoT stream computing systems. Due to the characteristics of real-time, volatile, emergent, disorderly, and infinite in data stream, data incompleteness issues such as data loss, duplication, and state inconsistency in stream computing systems are becoming more prominent, making the study of data integrity and consistency more difficult than ever [2].

Although most stream computing systems currently have an acker [3], which is a mechanism to check whether each message can be processed completely, the integrity of the message data itself is not guaranteed. At the same time, because the acker module is inside the system, the efficiency of real-time message computation is easily affected if complex validation computations are run on the acker. Since the stream computing process is not persistent, it is not possible to view the historical message processing path, making it difficult to reproduce the problem of incomplete message data.

This data security issue becomes a serious constraint to the application of stream computing systems in the Internet of Things.

In order to solve the above problems, this paper constructs an external data integrity tracking and verification system (i.e., external tracking and verification system) to monitor the integrity and correctness of message data content and processing path in the stream computing system, which can accurately record the processing path of each message and verify the integrity of message content, as well as detect errors, give an alarm, and recover messages in time without affecting the efficiency of the original stream computing system.

The external tracking and verification system satisfies the following capabilities:Accuracy: Accuracy is a key consideration for stream real-time computing systems used in IoT. Only with a high level of accuracy and precision can the system be trusted by end users and be widely applied.Real-time: Data sharing in IoT requires high timeliness; so, data integrity verification needs real-time. Since the tracking and verification system is built outside the stream computing system, integrity verification does not affect the efficiency of the original system. Meanwhile, the verification time is synchronized with stream computing, making it possible to trace and recover error messages as soon as possible.Transparency: Different stream computing systems for IoT may have different topological frameworks, corresponding to different business and application interfaces; thus, the design of external tracking and verification systems should be transparent in order to achieve system versatility.

This paper focuses on how to guarantee the integrity and consistency of message data stream, based on a generic stream computing system, using homomorphic message authentication code and pseudo-random function security assumption, constructing a data integrity verification algorithm scheme of the stream computing system (S-DIV), and building an external tracking and verification system to record and analyze the data content and processing path of each message in real time, proactively discover the incompleteness of message data, and automatically give an alarm and recovery, to guarantee the integrity and consistency of message data throughout the entire lifecycle from acquisition, transmission, computation, to storage. Then, we analyze the security of the S-DIV scheme under the standard model, and finally, simulate the real-time computation and verification process of S-DIV scheme; and provide the analysis in terms of preprocessing computational efficiency, proof generation efficiency, verification efficiency, timing analysis, storage cost and communication cost, respectively. The experimental results show that the S-DIV-based external tracking and verification system can efficiently guarantee the integrity of message data without affecting the original efficiency of the stream real-time computing system and replay the error data within an acceptable time range; thus, it can solve the problems of easy-loss and nonretrievability of message data stream.

## 2. Related Work

The concept of data integrity verification was first applied in grid computing and P2P network. Initially, a hash function was used to verify the integrity of remote data. The hash value was stored locally and the data were stored on the remote node. Then, the data were retrieved and the hash value was calculated at each verification and compared with the locally stored hash value. Therefore, the communication cost and computational cost of each verification are very high.

With the development of technology, data integrity verification schemes based on different cryptographic primitives and other technologies have been proposed.

Schemes based on RSA

Ateniese et al. [4] proposed a probabilistically secure PDP scheme using RSA scheme, homomorphic tags, and sampling detection, where a fixed number of data blocks are randomly selected and aggregated into a smaller value at each verification, thus greatly reducing the communication cost and computational cost. Erway et al. [5] proposed a Dynamic PDP (DPDP) scheme that can support data updates, which is based on the S-PDP model with a jump table structure to add, modify, and delete data. Wang et al. [6] proposed a nonrepudiable PDP scheme, which constructs a reusable commitment function scheme based on the Pederson commitment. This guarantees that the user cannot deny a correct proof when verifying the data; thus, it guarantees the security of both the user and the server. Chinnasamy et al. [7] introduced an improved key generation scheme for RSA to enhance the security of clients’ information, which can be applied to the construction of PDP and POR schemes.

2.Schemes based on symmetric encryption

Ateniese et al. [8] proposed an efficient and securely verifiable PDP scheme based on a symmetric cryptosystem, and the scheme can support dynamic manipulation of data, such as modification, addition, and deletion of data, but the number of verifications is limited. Juels et al. [9] proposed a POR scheme based on symmetric encryption to recover some errors while verifying data integrity. The POR scheme proves the integrity by adding “sentinels” to the encoded data, randomly selecting the sentinels for comparison, and by adding error-correcting codes to achieve data recoverability. Chang et al. [10] proposed a Remote Integrity Check (RIC) model, which is similar to the POR model and can remotely verify the integrity of the data; subsequently, a POR scheme for dynamic data updates has been designed on this basis.

3.Schemes based on homomorphic verification tags

Shacham et al. [11] proposed a CPOR mechanism using homomorphic verification tags and gave separate schemes for public and private verification, which greatly reduces the communication cost and solves the drawback of limited verification number of the POR scheme. Curtmola et al. [12] proposed the integrity verification scheme of data stored on multiple cloud servers, which combines the traditional error correction codes with the data possession problem. Wang et al. [13] used homomorphic tokens to verify the integrity of data stored on multiple servers, which can reduce the communication cost required under distributed storage. However, the scheme does not support dynamic data updates and it can easily be identified which server the data is stored on, leading to low security.

4.Schemes based on elliptic curve

Hanser et al. [14] proposed an efficient PDP scheme based on elliptic curves that can support both private and public verification, which uses the same preprocessing phase to generate verification tags that support both private and public verification. Wang et al. [15] proposed an efficient PDP scheme based on elliptic curves, which replaces the homomorphic multiplication of RSA with the homomorphic addition of elliptic curves and improves the computational efficiency.

5.Schemes based on bilinear mapping

Zhu et al. [16] constructed a Cooperative PDP (CPDP) scheme based on a hierarchical hash index structure and bilinear mapping for hybrid clouds. The scheme can support data migration and service scaling in both private and public cloud environments. However, the computational efficiency is relatively low due to the bilinear mapping.

6.Schemes based on the third party

Wang et al. [17,18] introduced trusted third party (TTP) into the data integrity verification scheme to protect the privacy of data and user identity, using TTP instead of client to verify the integrity of data, while TTP cannot obtain the specific data content during the verification process. However, TTP is difficult to implement in real applications and can cause additional overhead to the user. Armknecht et al. [19] proposed an outsourced POR (OPOR) model—Fortress, which uses an untrusted third party on behalf of the user to interact with the server; it can protect the user, the cloud server, and the third party simultaneously.

7.Schemes based on the group

Tate et al. [20] firstly proposed a group data integrity verification scheme based on trusted hardware for multiuser data sharing in cloud storage with data update support. Wang et al. [21] proposed a group scheme supporting identity privacy protection and constructed separate schemes supporting private and public authentication. The scheme ensures that the identity information of group members will not be leaked during the integrity verification process. Subsequently, a public authentication scheme was proposed to support revocation of group members [22], which ensures that revoked members cannot forge legitimate tag values or proofs using known information. Wang et al. [23] proposed a group PDP scheme supporting data deduplication. Based on the homomorphic message authentication code and the L-cross authentication code, the scheme ensures that group members with shared data can use the same tag to complete the verification and to achieve the purpose of tag data deduplication; further, since the verification keys of each member are independent, the scheme can resist selected member attacks. Zhu et al. [24] proposed a group provable of storage scheme with malicious-member distinction and revocation (DR-GPOS) based on the homomorphic authentication code, pseudo-random function, and commitment function; the scheme can distinguish malicious group members and prevent selected member attacks and collusion attacks.

8.Schemes based on other technologies

Chen et al. [25] proposed a scheme to solve the data integrity problem when data are stored on multiple servers in a cloud environment, using network coding instead of the previous fault-tolerant codes to achieve the goal of supporting dynamic data updates. Halevi et al. [26] proposed the concept of Proofs of Ownership (POW) to address the problem of data duplication attacks and permission attacks. Cao et al. [27] used LT codes in data possession to achieve distributed cloud-based data integrity verification, which can reduce the computational overhead of decoding and data calculation. Zheng et al. [28] proposed a POS scheme supporting data deduplication, which combines a POW scheme with a POS scheme, and reduces the number of tag values of duplicate data in public verification, thus reducing the storage cost of the server.

9.Schemes based on the Frontier emerging technologies

In recent years, with the development of technologies such as blockchain and biomedicine, data integrity verification has received new insights. Li et al. [29] proposed a data integrity audit structure based on fuzzy identity. By using biometric identification technology to construct an identity fuzzy mechanism, it solves the complex key management problem in cloud. This scheme has certain fault tolerance but requires large computational and communication cost. Guo et al. [30] proposed a batch update algorithm based on Merkle hash tree, which can perform and verify multiple update operations at one time and avoids repeated computation and transmission. Subsequently, a dynamic proof of data possession and replication (DPDPR) scheme is proposed [31], which stores the indexed Merkle hash tree based on the data block in the cloud to reduce the storage cost. However, when the data are uploaded, the cloud service provider needs to verify the hash value of each data block, leading to high computational cost, and the verification cost is expensive when used to support dynamic operations. Yaling et al. [32] introduced a multibranch tree to realize dynamic update, which simplifies the structure of verification, but the data copy and tag generation are all performed by the user, which increases the computational pressure of the user. Xu et al. [33] proposed a distributed and arbitrable remote data auditing scheme based on blockchain, using smart contracts to notarize the integrity of outsourced data on the blockchain, and using the blockchain network as a self-recording channel to achieve nonrepudiation verification. Li et al. [34] proposed a cloud-side collaborative stream data integrity verification model based on chameleon authentication tree, which uses a trusted third party to complete stream data insertion, query, integrity verification, and data confidentiality.

In summary, there are few researches on integrity verification of stream data currently, and most of the existing schemes do not meet the requirements of efficiency and real-time computing in IoT stream system; thus, it is necessary and meaningful to form a solution to verify the integrity of stream data in IoT.

## 3. Model Construction

### 3.1. Model Design

The overall architecture design of the external tracking and verification system of the stream computing system (improved and adapted from [35]) is shown in Figure 1.

The external tracking and verification system is tightly coupled with the stream real-time computing system, and the internal is mainly composed of five modules: message tracking data center, key management center, batch data preprocessing module, batch data verification module, and alarm module. The external tracking and verification system can provide an interface coupled with stream computing system for real-time data collection.

### 3.2. Overall Work Steps

The tracking and verification process can be divided into the following phases:The phase of real-time data collection: The external tracking and verification system has a data collection interface on each data-processing module of the stream computing system, which is used to record and send message data to the message tracking data center. Each message defines a unique message ID, which remains unchanged during all phases. When a message enters a module or leaves this module, a message state is defined to record information such as the receiving module or the sending module, and the two states are submitted to the tracking data center.The phase of message classification: in the message tracking data center, each message is classified and preprocessed with the message ID as the unique identifier, and all the intermediate data of this message are merged to form a batch of message data.The phase of key generation: the key management center distributes the pregenerated key (symmetric key or public–private key pair) to the data preprocessing module and data verification module.The phase of batch data preprocessing: The system uses cryptographic algorithms to preprocess the batch data of each message—that is, it uses the key to generate verification tags. Specifically, the system computes the message data sent by each data-processing module in turn, and, respectively, generates verification tags to verify the integrity of the received message data of the next data-processing module.The phase of batch data integrity verification: Use the corresponding algorithm, verification tag, and key to verify the integrity of each message batch data. Specifically, each receiving module verifies the integrity of the received data according to the verification tag, key, and message status (i.e., sending module and receiving module information, message ID, etc.).The phase of alarm and recovery: If the verification passes, it means that the message has been completely and correctly processed; then, the related intermediate data are deleted. If the verification fails, it means that the data integrity is abnormal and the message is not processed correctly; then, alarm and recovery phase is executed, which checks the records of this error message- in the external tracking and verification system, resends the message to the message queue, and recalculates and reprocesses it in the stream system.

## 4. Preliminaries

### 4.1. Homomorphic Message Authentication Code

Homomorphic message authentication code [36] is widely used in network coding, and its formal definition is as follows:

**Definition** **1.***The homomorphic message authentication code (HomMac) scheme consists of four probabilistic polynomial time algorithms* (Gen,Sign,Combine,Verify)*:*

$Gen(1^{\lambda })\to Key$

*is a probabilistic algorithm to generate the key Key; it inputs the security parameter*

λ

*and outputs the secret key Key.*
$Sign(Key,id,v_{i},i)\to t_{i}$*is an algorithm to generate a tag value, it inputs key Key, vector space identifier id, vector value*$v_{i}\in F_{q}^{s}$*, and vector index value i, and outputs the corresponding tag value*$t_{i}$.$Combine((v_{1},t_{1},\alpha_{1}),\ldots ,(v_{c},t_{c},\alpha_{c}))\to (T,y)$*is a probabilistic algorithm to generate HomMac, it inputs c random constant parameters*$\alpha_{1},\ldots ,\alpha_{c}\in F_{q}$*, c vectors*$v_{1},\ldots ,v_{c}\in F_{q}^{s}$*, and corresponding tag values*$t_{1},\ldots ,t_{c}$*; calculates the corresponding HomMac value*$T=\sum_{i=1}^{c}\alpha_{i}t_{i}\in F_{q}$*and**aggregated value*$y=\sum_{i=1}^{c}\alpha_{i}v_{i}$*; and finally, outputs T and*y.

Verify(Key,id,y,T)→{1,0}

*is a deterministic algorithm to verify the HomMac value, it inputs the key Key, the vector space identifier id, the vector*

$y\in F_{q}^{s}$

*, and the corresponding T; if the HomMac value is correct, output 1; otherwise, output 0.*

*HomMac must satisfy the following two properties:*
***Correctness***: *For the vector space identifier id, vectors*$v_{1},\ldots ,v_{m}\in F_{q}^{s}$, *and the index value*i=1,…,m, *when*Key∈κ*, *$t_{i}=Sign(Key,id,v_{i},i)$, *and*$\alpha_{1},\ldots ,\alpha_{m}\in F_{q}$*, the probability of validation failure is negligible:*(1)$${\mathrm{fail}}_{\mathrm{HomMac}}(\lambda ):=\mathrm{Pr}[Verify(Key,id,\sum_{i=1}^{m}\alpha_{i}v_{i},Combine((v_{1},t_{1},\alpha_{1}),\ldots ,(v_{m},t_{m},\alpha_{m})))\neq 1]\leq negl(\lambda )$$***Security***: *Choosing the message attack of HomMac is defined as follows: Assume that the adversary has the ability to select message query; the adversary can obtain the tag values of a series of vectors adaptively, for the vector space*$V_{i}$*with unique identifier*$id_{i}$, *if the adversary can generate a valid triple*(id,y,t)*, where id is the new identifier or*$id=id_{i}\wedge y\notin V_{i}$*, it indicates that the adversary can successfully forge a HomMac, and the probability of such forgery is negligible.*

### 4.2. Pseudo-Random Function

The informal definition of the pseudo-random function (PRF) [37] is as follows:

**Definition** **2**.*Assume that*$I_{k}$*is the set of bit strings of length k, and*$H_{k}$*is the set of all functions mapped from*$I_{k}$*to*$I_{k}$. *Then, the pseudo-random function*$F=F_{k}\subseteq H_{k}$*satisfies the following properties:****Indexability***: *Each function has a unique index value k; according to k, a function*$f\in F_{k}$*can be easily, randomly selected from the set.****Polynomial-time****: Given an input x and a function*$f\in F_{k}$*, there exists a probabilistic polynomial-time algorithm that can easily solve for the value of*f(x).***Pseudo-randomness****: There is no probabilistic polynomial-time algorithm that can distinguish a function in*$F_{k}$*from a function in*$H_{k}$*, nor can it distinguish a value of*f(x)*from a random number in*$I_{k}$.

### 4.3. Negligible Function

Negligible function [38] is a concept in modern cryptography based on computational complexity theory, which is used to prove that security technology is mathematically provably secure; it is defined as follows:

**Definition** **3**.*For a function*μ(x):N→R, *if for any positive polynomial*poly(x), *there exists a*Nc>0, *such that for all*x>Nc, *satisfies*(2)$$\mu (x)<\frac{1}{poly(x)}$$*then,*μ(x)*is a negligible function, generally expressed as*negl(x).
*In general,*

negl(x)

*is equal to the polynomial function of x divided by the exponential function of x, so that as the value of x tends to infinity, the value of*

negl(x)

*is infinitely close to 0.*


## 5. Implementation

### 5.1. Scheme Implementation

#### 5.1.1. Formal Definition of Data Integrity Verification Scheme of Stream System

**Definition** **4**.
*A Data Integrity Verification Scheme of the Stream Computing System (S-DIV) consists of four polynomial-time algorithms*

(KeyGen,PreGen,ProofGen,VerifProof)

*:*


$KeyGen(1^{\lambda })\to Key$

*is a probabilistic algorithm to generate key, it inputs security parameter*

λ

*and outputs the secret key Key.*
$PreGen(Key,M_{\left(Mid,Sid\right)})\to T_{\left(Mid,Sid\right)}$*is an algorithm to generate verification tag; it inputs the key Key and message data*$M_{\left(Mid,Sid\right)}$, *and generates the tag value*$T_{\left(Mid,Sid\right)}$*of the message data.*$ProofGen(Key,{\left\{M\right\}}_{Mid},{\left\{T\right\}}_{Mid})\to \rho$*is an algorithm to generate a proof of data integrity, it inputs the secret key Key, the message data set*${\left\{M\right\}}_{Mid}$*identified by Mid, and the corresponding tag set*${\left\{T\right\}}_{Mid}$*, and generates a data integrity proof*ρ*, which can be regarded as the data aggregation of*${\left\{M\right\}}_{Mid}$*and*${\left\{T\right\}}_{Mid}$.VerifProof(Key,ρ)→{1,0}*is a deterministic algorithm to verify the data integrity proof; it inputs Key and the proof*ρ, *and outputs whether*ρ*is a correct proof.*

#### 5.1.2. A Structure of Data Integrity Verification Scheme of Stream System

Let $f:\kappa \times I\to F_{q}$ be a pseudo-random function, and *q* be the order of the finite field $F_{q}$, whose size depends on the security parameters λ (for example, $q=2^{\lambda }$). Then, the key space of S-DIV is $\kappa =F_{q}^{3}$ and the tag space is $F_{q}\cup \{⊥\}$, where ⊥ means no solution or multiple solutions.
$KeyGen(1^{\lambda })\to (a,b,c)$

Input the security parameter λ, and output key (a,b,c)∈κ, where a,b,c is the value randomly selected in $F_{q}$.




$PreGen((a,b,c),M_{\left(Mid,Sid_{i}\right)})\to T_{\left(Mid,Sid_{i}\right)}$




Calculate the tag equation: (3)$$a\cdot M_{\left(Mid,Sid_{i}\right)}+f_{b}(Mid||Sid_{i})=c\cdot T_{\left(Mid,Sid_{i}\right)}$$
where || is the splicing, and the tag value $T_{\left(Mid,Sid_{i}\right)}$ can be efficiently calculated by solving the one-dimensional linear equation.

Finally, output the tag $T_{\left(Mid,Sid_{i}\right)}$.




$ProofGen(Key,{\left\{M\right\}}_{Mid},{\left\{T\right\}}_{Mid})\to \rho$




Aggregate message according to all the session messages (i.e., all messages with the same *Mid* and different *Sid*), and aggregate tag according to all the tag values (i.e., all tags with the same *Mid* and different *Sid*), to generate integrity proof.

Assume the number of session messages is *z*, select a random value $k\in F_{q}$, for 1≤j≤z:

Compute random parameter for each session message: vj=fk(j).

Then, calculate message aggregate value and tag aggregate value, respectively:(4)$$\omega_{Mid}=v_{1}M_{Mid,Sid_{1}}+v_{2}M_{Mid,Sid_{2}}+\ldots +v_{z}M_{Mid,Sid_{z}}$$
(5)$$\tau_{Mid}=v_{1}T_{Mid,Sid_{1}}+v_{2}T_{Mid,Sid_{2}}+\ldots +v_{z}T_{Mid,Sid_{z}}$$

Finally, output vector $\rho =(\omega_{Mid},\tau_{Mid})$.




VerifProof(Key,ρ)→{1,0}




Calculate and verify the message aggregate value of the receiving module and the tag aggregate value of the sending module:

For 1≤j≤z:

Compute random parameter for each session message: $v_{j}=f_{k}(j)$.

The verification process is as follows:(6)$$\sigma_{Mid}=v_{1}f_{b}(Mid||Sid_{1})+v_{2}f_{b}(Mid||Sid_{2})+\ldots +v_{z}f_{b}(Mid||Sid_{z})$$

Verify whether the equation $a\cdot \omega_{Mid}+\sigma_{Mid}=c\cdot \tau_{Mid}$ is correct; if so, output 1; otherwise, output 0.


**Notice:**
The identifier *Mid* is randomly generated by a pseudo-random function when the message is generated. As the unique identifier of the message, it is used to distinguish messages with the same content but that are actually different. *Mid* remains unchanged throughout the life cycle of the message; thus, it can effectively resist collision in an actual system. After the stream system completes all calculations of a message and the integrity verification is correct in the entire cycle, the message data are deleted together with *Mid*.The identifier *Sid* is generated when the message is transmitted between modules. The *Sid* of each session (transmission from one module to another is called a session) is different. It is used to distinguish the message with the same content but that is actually different or the message in different processing phase but with same content. After the stream system completes all calculations of a message and the integrity verification is correct in the entire cycle, the message data are deleted together with *Mid*.The algorithm ***ProofGen*** can be regarded as an aggregation function, which is used to aggregate all the different messages of the same *Mid* collected by the receiving module and perform batch verification according to verification tags with corresponding *Mid* and *Sid* generated by the sending module. It improves computational efficiency.


### 5.2. System Detailed Design

According to the S-DIV scheme, construct the external data integrity tracking and verification system as follows.

#### 5.2.1. Message Format Design

First, the detailed design of the message format is introduced. Each message $M_{\left(Mid,Sid\right)}$ is composed of four parts: the sending and receiving flag (*Flag*), the message ID (*Mid*), the session ID (*Sid*), and the message content data (*M*). The formal expression is $M_{\left(Mid,Sid\right)}=(Flag||Mid||Sid||M)$, where || is splicing.

Concretely, Flag is 2 bits, which is used to indicate whether the message is sent or received: if *Flag = 00*, it means that the message is collected by the sending port and is a sending message. If *Flag = 01*, it means that the message is collected by the receiving port and is a received message. If it is other, it means that the message is incorrect and needs to be recollected.

*Mid* is 32 bits, which is the unique identifier of the message itself. In the actual system, the *Mid* of a message is calculated by a pseudo-random function. When the message is correctly verified, the message and the identifier will be deleted together. In this case, 32-bit random number can satisfy the uniqueness of the message ID in most systems.

*Sid* is 32 bits, which is the session identifier of the message transmitted in different modules. When the message is in module A, module A calculates and processes the message; when it sends the message, it modifies the *Sid*, generates a new message, and sends it to the next module B. By repeating this, the session identifier *Sid* in the session transmission between modules is different. Since the *Sid* of the sending port of module A and the receiving port of module B are the same, it is used to verify that the message has not been tampered with or lost during transmission.

#### 5.2.2. Detailed Design of External Data Integrity Tracking and Verification System

The detailed design of the external data integrity tracking and verification system of stream computing system is shown in Figure 2.
The phase of real-time data collection: When a message is sent from module A to module B, data collection is performed at the data sending port (module A) and the data receiving port (module B), and the collected message data are sent to the message tracking data center.The phase of message classification: After the data center receives the collected message, it judges whether it is a sending message or a receiving message according to the Flag, the sending message (*Flag = 00*) will be put into the sending data storage module (i.e., sending module), and the receiving message (*Flag = 01*) will be put into the receiving data storage module (i.e., receiving module).The phase of key generation: The key management center sends the pregenerated key to the batch data preprocessing module and the batch data verification module, respectively.The phase of batch data preprocessing: The preprocessing module preprocesses the message data of sending module, calculates each message $M_{\left(Mid_{1},Sid_{1}\right)}$, and generates a verification tag $T_{\left(Mid_{1},Sid_{1}\right)}$; then, it sends the tag to the batch data verification module.The phase of batch data integrity verification: The messages of receiving module are sent to batch data verification module. The batch data verification module verifies data integrity of the message one by one according to $T_{\left(Mid_{1},Sid_{i}\right)}$ and $M_{\left(Mid_{1},Sid_{i}\right)}$. Specifically, aggregate $T_{\left(Mid_{1},Sid_{i}\right)}$ and $M_{\left(Mid_{1},Sid_{i}\right)}$ according to *Mid*: aggregate and verify a set of messages ${\left\{M\right\}}_{Mid_{1}}=\{M_{\left(Mid_{1},Sid_{1}\right)},M_{\left(Mid_{1},Sid_{2}\right)},M_{\left(Mid_{1},Sid_{3}\right)},\ldots \}$ with the same *Mid* and the corresponding series of tags ${\left\{T\right\}}_{Mid_{1}}=\{T_{\left(Mid_{1},Sid_{1}\right)},T_{\left(Mid_{1},Sid_{2}\right)},T_{\left(Mid_{1},Sid_{3}\right)},\ldots \}$. If the verification passes, the information will be sent to the message tracking data center and the stream computing system, and the intermediate data in the two caches will be deleted; if the verification fails, the message alarm and recovery will be carried out.The phase of alarm and recovery: When the alarm module receives the error information, it calls out the error message from the batch data verification module and resends the error message to the message tracking data center according to the *Mid* and *Sid*. The data center finds out the original message and sends it to the stream computing system. Finally, the stream computing system replays and recalculates the message according to the original route.

## 6. Security

**Theorem** **1.***Assume that f is a secure pseudo-random function; the S-DIV scheme is a secure data integrity verification scheme under the standard model*.

**Proof of Theorem** **1.**
The proof of correctness
The proof of the correctness of S-DIV scheme is essentially the verification of the legal proof:When $Key=(a,b,c)\leftarrow KeyGen(1^{\lambda })$, for any valid proof, the probability of verification failure is
(7)$${\mathrm{fail}}_{S-\mathrm{DIV}}(\lambda ):=\mathrm{Pr}[VerifProof(Key,ProofGen({\left\{M\right\}}_{Mid},{\left\{T\right\}}_{Mid}))\neq 1]$$The equation for tag calculation can be expressed as $\tau =c^{-1}(a\cdot \omega +\sigma )$, where τ is the aggregation of tag values, *a* and *c* are randomly generated keys, $c^{-1}$ is the inverse of *c*, ω is the aggregation of messages, and σ is the aggregation of pseudo-random function values. ***VerifProof*** will fail to verify when one-dimensional linear equation of tag calculation has no solution or multiple solutions.The probability can be obtained as ${\mathrm{fail}}_{S-\mathrm{DIV}}(\lambda )\leq \frac{1}{2^{\lambda }}$.
2.The proof of reliability
The proof of the reliability of the S-DIV scheme is essentially the verification of the illegal proof:For any illegal proof $\rho^{\prime }\notin \{ProofGen({\left\{M\right\}}_{Mid},{\left\{T\right\}}_{Mid})\}$, the probability of passing the verification is
(8)$${\mathrm{Adv}}_{{}_{S-\mathrm{DIV}}}^{illeg}(\lambda ):=\underset{Key}{\max}\mathrm{Pr}[VerifProof(Key,\rho^{\prime })=1|Key\leftarrow KeyGen(1^{\lambda })]$$
where *max* refers to traversing all *Key*, and the probability of passing the verification is equal to the probability of finding *n* random numbers to satisfy ***VerifProof***.The probability can be obtained as ${\mathrm{Adv}}_{{}_{S-\mathrm{DIV}}}^{\mathrm{illeg}}(\lambda )\leq \frac{1}{q^{n}}$, where $q=2^{\lambda }$ and *n* is the total number of message data.
3.The proof of data integrity
The parameters are simplified when proving data integrity; set all parameters $v_{j}$ to 1, all subscripts are omitted. The pseudo-random function *f* and random number *r* are used to execute data integrity scheme in a real and ideal environment, respectively.
**In real environment:**
**Setting**: The challenger runs $KeyGen(1^{\lambda })$ to generate the key *Key* and keep it private.**Query**: The adversary selects the message data *M* and sends to the challenger; then, the challenger runs PreGen(Key,M) to calculate the tag value *T*, whose process is as follows: make $T=c^{-1}aM+c^{-1}B$, where *a* and *c* are the keys and $c^{-1}$ is the inverse of *c*, calculate and send *T* to the adversary, and this query can be performed many times. Then, the adversary stores all message data set {*M*} and tag value set {*T*}.**Challenge**: The challenger requests the integrity proof of all message data blocks {*M*} from the adversary.**Forgery**: The adversary calculates an integrity proof ρ and sends it to the challenger.
**In ideal environment:**
Use the random number instead of pseudo-random function—that is, *B = r*, where *r* is a pure random number.If the adversary can complete the verification under the condition that *M* is changed, in other words, in the ideal environment, the adversary can successfully find $T^{\prime }$ to satisfy the equation $T^{\prime }=c^{-1}aM^{\prime }+c^{-1}B$, where $M^{\prime }$ is the changed *M*.Since *a* and *c* are randomly generated keys and *B* is a pure random number, the probability of successful forgery is as follows:(9)$$A_{S-\mathrm{DIV}}^{\mathrm{ideal}}=\mathrm{Pr}[T^{\prime }|T^{\prime }=c^{-1}aM^{\prime }+c^{-1}B]=\mathrm{Pr}[T^{\prime }|T^{\prime }=r_{1}M^{\prime }+r_{2}]\leq \frac{1}{2^{\lambda }}$$
where $r_{1},r_{2}$ are random numbers.According to the assumption, *f* is a secure pseudo-random function; then, the adversary cannot distinguish whether the scheme is executed in the real environment or in the ideal environment.Therefore, in the real environment, the probability of adversary forgery, $A_{S-\mathrm{DIV}}^{\mathrm{real}}≅A_{S-\mathrm{DIV}}^{\mathrm{ideal}}\leq \frac{1}{2^{\lambda }}$, is negligible.Proof completed. □

## 7. Experiment and Analysis

Experimental environment: All data are stored in Kafka message queue (version 2.8.1) and calculated in real time in storm framework simulation environment, and the cryptographic library is OpenSSL (version 0.9.7a). All experiments are conducted on a machine designed for data-intensive processing, each compute node has 128 GB of memory and 12 cores in docker container. Since the results of each experiment will be slightly different due to environmental deviation, all data results are obtained by comparing and averaging for multiple tests.

The verification in storage mode mostly adopts the sampling method, but this experiment uses all the message data for calculation, and the verification accuracy can reach 100%.

### 7.1. Comparison of Various Schemes

The comparison of various data integrity verification schemes is shown in Table 1, where *n* is the total number of data blocks or messages, *c* is the number of sampled data blocks in a proof, *w* is additional storage such as tags, and *l* is the number of data block splits.

As can be seen from Table 1, even though the computational complexity is the same, S-DIV scheme only uses multiplication and addition operations, which is much more efficient than the exponentiation and multiplication operations of other schemes based on BM, EC, RSA, etc. At the same time, the communication and storage costs of S-DIV scheme are smaller. Thus, the S-DIV scheme is more suitable for real-time stream computing systems than other schemes.

### 7.2. Computational Efficiency Comparison under Different Message Concurrency

In the S-DIV scheme, each message’s data are corresponding to a verification tag. In the message collection phase, the data center determines the storage queue according to the value of *Flag*; then, the preprocessing module processes the messages in the storage pool of *Flag = 00*: according to two identifiers *Mid* and *Sid* of each message, calculate the corresponding tag value. Finally, the verification module selects the corresponding message data from the storage pool of *Flag = 01* according to identifiers *Mid* and *Sid*, and aggregates and verifies the integrity of messages based on tag values.

Figure 3 shows the performance of the S-DIV scheme under different message concurrency (improved and adapted from [35]).

The time of the data collection phase is completely determined by the time of different message data generation in the original stream computing system and has no reference for the performance of the S-DIV scheme; so, the data collection time is ignored. The preprocessing time only records the time to calculate the verification tag, ignoring the times of message data retrieval, lookup, reading, and transmission—these times are collectively referred to as I/O time. Verification time records the time to aggregate and verify the data and tags of all messages, ignoring the I/O time. The message alarm and recovery time depends on the efficiency of the original stream computing system; so, it is not considered in the performance experiment of the S-DIV scheme.

It can be seen from Figure 3 that as the number of messages increases, the preprocessing time (PreGen), proof generation time (ProofGen), and verification time (VerifProof) all increase linearly; as the S-DIV scheme is different from the previous sampling verification scheme, it preprocesses and verifies all messages so that as the amount of message data increases, the calculation amount of all phases increases linearly. However, when the message concurrency is small (below 100), the calculation time of preprocessing, proof generation, and verification are in milliseconds; when the concurrency of messages is larger (1000), the calculation time of preprocessing, proof generation, and verification are all about 0.1s, which satisfies the real-time performance of stream system. Simultaneously, because the S-DIV scheme is an external subsystem of the stream system, it has little impact on the internal performance of the stream real-time computing system.

### 7.3. Comparison of Preprocessing Efficiency between S-DIV and Traditional PDP

Since there is no relatively authoritative data integrity verification scheme in the current stream computing system, we selected the authoritative integrity verification scheme in batch computing mode—PDP as the reference scheme. The PDP scheme is a probabilistic security scheme proposed by Ateniese et al. [4], which uses the RSA scheme, homomorphic tags, and sampling detection method. It is the most classic scheme in the field of data integrity verification, and many subsequent schemes are improvements and functional additions based on the PDP scheme. In the experiment, the S-DIV scheme and the PDP scheme are used to preprocess the same amount of message data, and the calculation time is recorded.

Figure 4 shows the calculation efficiency comparison between the S-DIV scheme and the PDP scheme in the preprocessing phase under the condition of different numbers of messages. In the experiment, only the calculation time during preprocessing is recorded, while other I/O times such as reading and searching are ignored.

As can be seen from Figure 4, on one hand, with the increase in the number of messages, the calculation time increases linearly. This is because in the preprocessing phase, both the S-DIV scheme and the PDP scheme preprocess all the message data and calculate to generate the verification tag; so, the amount of computation increases linearly with the amount of message data. On the other hand, the computation time required by the S-DIV scheme is generally less than that of the PDP scheme, because the PDP scheme is constructed based on the RSA signature scheme, which uses exponentiation and multiplication calculation, and requires relatively large calculation complexity, while the S-DIV scheme is constructed based on the homomorphic message authentication code and pseudo-random function, which uses multiplication and addition calculation, and requires lesser calculation complexity. Therefore, as the amount of message concurrency increases, the computational efficiency advantage is more obvious.

### 7.4. Comparison of Verification Efficiency between S-DIV and Traditional PDP

Figure 5a,b shows the comparison of the computational efficiency of proof generation and proof verification in the verification process of S-DIV and PDP, respectively. In the traditional PDP scheme, sampling calculation is used in the verification process, and a fixed number of data blocks are randomly selected for calculation and verification each time. For the fairness of the experiment, in the proof generation and verification phase, the S-DIV scheme and the PDP scheme select the same number of message data (the total number of messages) to generate proof and verify the proof. To intuitively compare the efficiency, only the calculation time during generating proof and verification is recorded in the experiment, and other I/O times are ignored.

It can be seen from Figure 5a that because all message data and verification tags are used to generate proof, the computation time for proof generation increases linearly. Meanwhile, because the PDP scheme uses exponential and multiplication operations, while the S-DIV scheme uses multiplication and addition operations, the computational complexity of the S-DIV scheme is lower than that of the PDP scheme.

It can be seen from Figure 5b—which is similar to the proof generation phase—that because all messages are used for verification, the calculation time increases with the increase in the message number. Further, due to differences in computational operations, the S-DIV scheme has lower computational complexity, and the computational efficiency is significantly improved.

### 7.5. The Timing Analysis

Figure 6a,b, respectively, show the timing diagram of one message and multiple messages in S-DIV scheme. The processing of each message includes three phases: preprocessing, proof generation, and proof verification. In the experiment, the calculation time of multiple randomly selected messages is collected and compared.

As can be seen from Figure 6, the calculation time of one message (including each processing phase) is at the millisecond level. Although the calculation times of multiple randomly selected messages are different from each other, all calculation times are within an acceptable range, which is suitable for the integrity verification of real-time data stream.

### 7.6. Storage Cost and Communication Cost

#### 7.6.1. Storage Cost

In the PDP scheme, the tag length of each data block is the same, which only depends on the security parameter and has nothing to do with the size of the data block. If the file block size is 4 KB and the modulus *N* is 1024 bits, the additional storage space is 3.125% of the original file data volume, and in the traditional storage mode, the operation method is to store firstly and then calculate; so, the data need to be stored for a long time, and the verification tag is also stored with the corresponding file data, meaning the storage overhead is greater.

The storage cost required by the S-DIV scheme is relatively small; the size of the storage space required by each tag is only related to the security parameter and has nothing to do with the length of the message. When the message stream is calculated completely and the verification result is correct in the external tracking and verification system, the message stream will be deleted in the cache and will not occupy additional storage space.

#### 7.6.2. Communication Cost

In the data collection phase, the communication bandwidth is equal to the total amount of message data, and the communication cost is O(n). In the preprocessing and verification phases, all computing operations are performed inside the external tracking and verification system, so there is no data transmission cost. When an error is found in the alarm and recovery phase, it only needs to resend the original message to the message buffer pool of the stream computing system, and the communication cost is O(1).

## 8. Conclusions

This paper proposed a data integrity verification scheme of the stream computing system (S-DIV scheme), and further constructed an external data integrity tracking and verification system based on the stream computing system in IoT, to analyze and verify the integrity of each message data in real time. Then, the security analysis of the scheme is given under the standard model. The experimental simulation and result analysis are carried out in the stream computing system, and the experiment proves that the external tracking and verification system can effectively ensure the integrity of the message data without affecting the efficiency of the original system.

In future research, the main difficulties that need to be considered are as follows: (1) In the stream real-time computing system, there is a situation where messages are reasonably discarded, and the tracking and verification system needs to be able to distinguish this situation without generating a false alarm. (2) Currently, there are few authoritative papers or patents in this field, and further research and analysis in the promotion and application scenarios of IoT are needed.

## Figures and Tables

**Figure 1 sensors-22-06496-f001:**
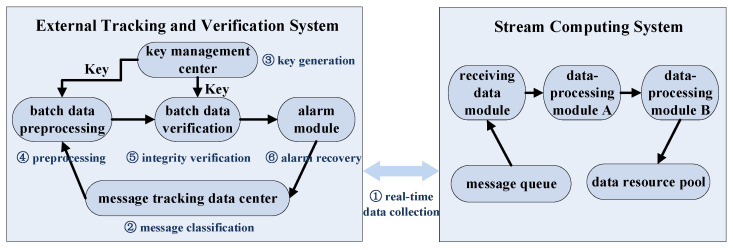
The architecture of external tracking and verification system of stream computing system.

**Figure 2 sensors-22-06496-f002:**
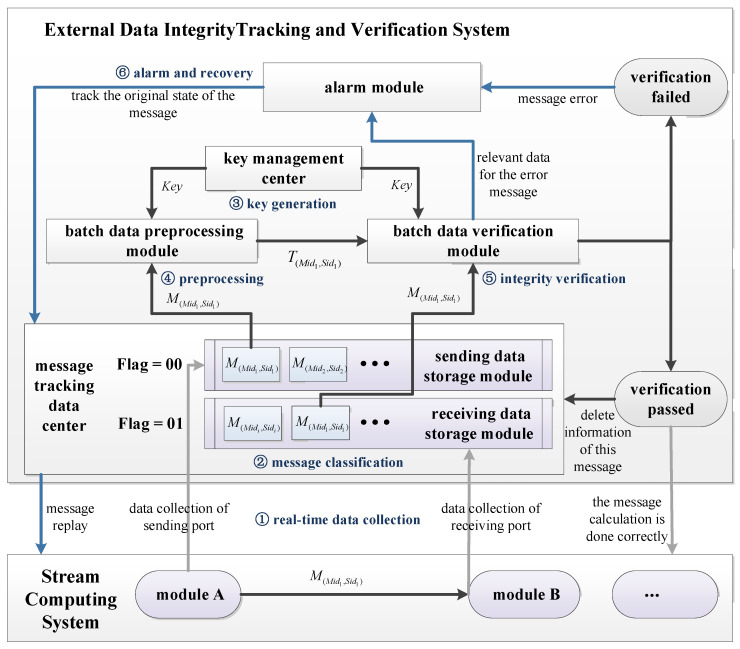
The detailed design of the external data integrity tracking and verification system.

**Figure 3 sensors-22-06496-f003:**
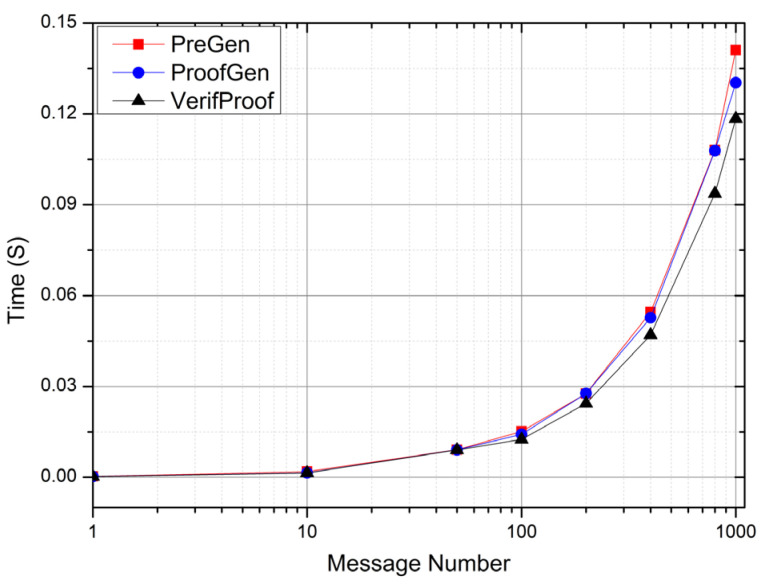
Overall performance of S-DIV scheme.

**Figure 4 sensors-22-06496-f004:**
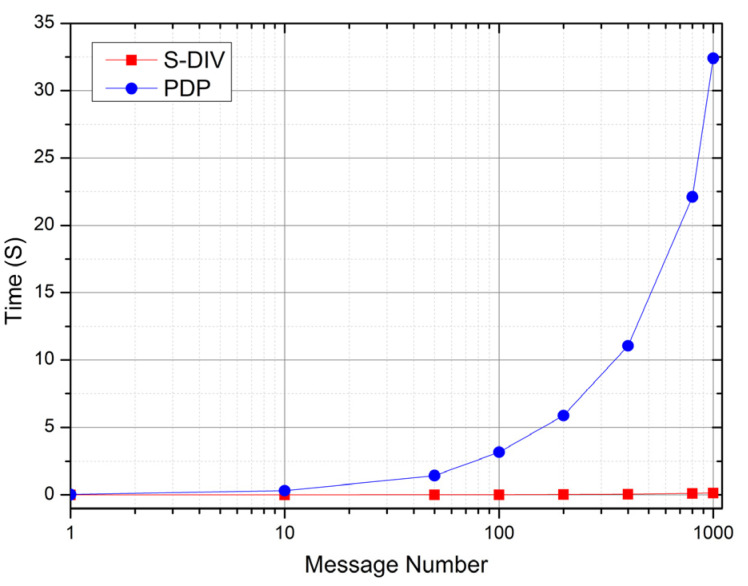
Comparison of preprocessing efficiency between S-DIV and PDP.

**Figure 5 sensors-22-06496-f005:**
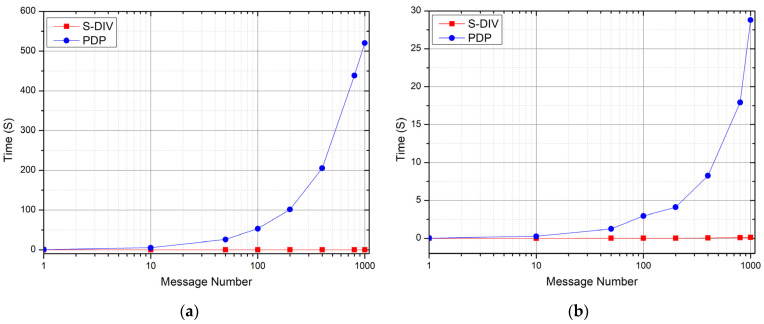
Comparison of verification efficiency between S-DIV and PDP: (**a**) comparison of proof generation; (**b**) comparison of verification.

**Figure 6 sensors-22-06496-f006:**
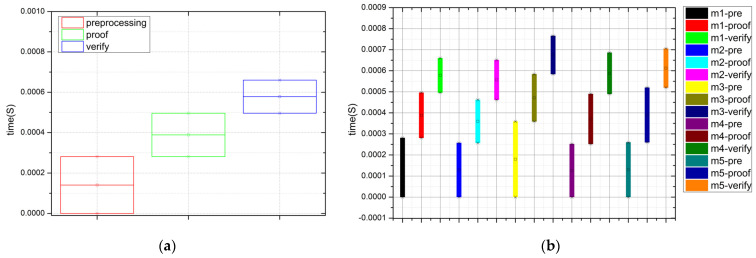
The timing diagram of S-DIV: (**a**) timing diagram of one message; (**b**) timing diagram of multiple messages (in the figure, m1 indicates message 1).

**Table 1 sensors-22-06496-t001:** The comparison of various data integrity verification schemes.

	S-PDP Based on RSA [4]	POR Based on SE [9]	CPOR Based on HVT [11]	E-PDP Based on EC [15]	CPDP Based on BM [16]	OPOR Based on TP [19]	S-DIV
Data integrity	Yes	Yes	Yes	Yes	Yes	Yes	Yes
Sampling verification	Yes	Yes	Yes	Yes	Yes	Yes	No
Preprocessing	O(n)	O(n)	O(tlogn)	O(n)	O(tlogn)	2O(n)	O(n)
Proof generation	O(c)	O(c)	O(c)	O(c)	O(c)	O(c)	O(n)
Proof verification	O(1)	O(1)	O(1)	O(1)	O(1)	O(1)	O(1)
Communication	O(1)	O(1)	O(l)	O(1)	O(l)	2O(l)	O(1)
Storage	O(n+w)	O(n+w)	O(n+w)	O(n+w)	O(n+w)	O(n+w)	O(1)
Real-time	No	No	No	No	No	No	Yes
Data stream	No	No	No	No	No	No	Yes

SE, HVT, EC, BM, and TP are the abbreviations of symmetric encryption, homomorphic verification tags, elliptic curve, bilinear mapping, and third party, respectively.

## Data Availability

The data for this study are available from the author upon reasonable request.

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
