# Peer review of "On the Design and Implementation of the External Data Integrity Tracking and Verification System for Stream Computing System in IoT†"

_sensors, 2022, doi:10.3390/s22176496_

Round 1

Reviewer 1 Report

1. Keywords not appropriate. Update it

2. Equations numbering is not available

3. Validation of the Results is not available.

4. Results obtained must be validated with different parameters. 

5. After the introduction, add the related works and then proposed work followed by the results.

Author Response

We thank you for your valuable suggestions and comments, please see the attachment for point-by-point response.

Reviewer 2 Report

The author did research to design and implement a secure data integrity verification method for stream computing in IoT. I have some suggestions like

1. The Flow number is missing in the proposed architecture

2. The author has used so many symbols and equations, so it's advised to use Notations Table in the article. 

3. System settings are missing in the parameter setting section

4. The proposed method-based Use case study is missed in the article.

5. The Timing analysis is missing in the article

6. The proposed verification and Integrity checking method should compare with the recent methods like - https://doi.org/10.3390/math10010068.

Author Response

(The authors gave the same response as above.)

Reviewer 3 Report

The paper covers a very important topic related to data flow in various environments where IoT is used.
The issue is important because of the increasing computerization in very many fields. The proposed solution contributes to research in the area of data transfer integrity and efficiency.

Author Response

(The authors gave the same response as above.)

Reviewer 4 Report

Comments and Suggestions for Authors

 The manuscript gives a detailed overview of a tracking and verification solution for stream computing systems designed for IoT applications.

 Remark: General

The authors confirm that this manuscript is an extension of a previously published manuscript. Parts of the published manuscript are referenced and adjusted and improved in the actual manuscript.

BUT, a large part of the manuscript is a reprint of the mentioned conference paper [1] "Wang, H. An External Data Integrity Tracking and Verification System for Universal Stream Computing System Framework", with high identical text copy and sentence structures. Only the wording has been changed. This is not an interpretation; this is a copy.

I have read the mentioned conference paper [1], which required additional effort from me. In this way I was able to identify the identical parts in both manuscripts.

An interpretation of a pre-existing manuscript is useful to the reader since he does not have to read the referenced manuscript. But it is important to reproduce the referenced manuscript in your own words.

 Example:

·         Parts of chapter 2

·         Parts of chapter 4 - line 205 – 290 à see [1] page 33, 34 and 35

·         Line 366 – 381 à see [1] page 35 and 36

·         Line 392 – 422 à see [1] page 36

·         Line 377 – 495 à see [1] page 36

 Remark: Line 60:

Reading the manuscript required sufficient knowledge of the thematic from the reader, for example, explain "acker" in detail to help the reader for a better understanding of the manuscript content.  For example: Add a notice that “acker” is a type of message acknowledgment mechanism.

 Remark: Line 423 – 474

Combine the figures 4 and Figure 5b in one diagram. Explain the reasons for the different computing times only once and not three times with approximately the same text. This makes the manuscript easier to read.

 Remark: Line 496 – 601

The chapter 7 “Related Work” is a “Research Background” and should be moved after chapter 1 “Introduction”. The two-page chapter has only three paragraphs, which makes reading tedious. Please structure the chapter more with paragraphs.

 References are checked only randomly. All selected references found; some are not open access.

 The English language is appropriate and understandable. In my opinion, some sentences are unfavorable worded, which sometimes disrupts the flow of reading.

Example: Line 427 – 428

“… which uses the RSA scheme, homomorphic verification tag and sampling …

Better readability:

“… which uses the RSA scheme, a homomorphic verification tag and sampling …

In my view, the manuscript needs to be completely reorganized. Using an existing manuscript as the main reference is ok but reproducing content from other manuscripts must be in your own words.

I'm sorry, but in my opinion, I reject the manuscript.

Author Response

(The authors gave the same response as above.)

Round 2

Reviewer 2 Report

After a revision, the article contents are improved a lot now. Still, I have small suggestions, 

1. In section 2, schemas based on RSA algorithms, the author should discuss this article https://doi.org/10.1007/978-981-10-7200-0_31. 

Author Response

(The authors gave the same response as above.)

Reviewer 4 Report

Dear Authors

The manuscript has been extensively revised and expanded. The content is well presented and the results are clearly recorded.

The reuse of parts of the text from [1] has been better integrated into the manuscript.

Chapter 7 with the reference to Chapter 2 can be removed from the manuscript.

With these changes, the manuscript can be released in my opinion.

Author Response

(The authors gave the same response as above.)
